# Single-molecule photoreaction quantitation through intraparticle-surface energy transfer (i-SET) spectroscopy

Jian Zhou[1], Changyu Li[1], Denghao Li[1], Xiaofeng Liu[1], Zhao Mu[2], Weibo Gao [2], Jianrong Qiu [3] & Renren Deng [1,4✉]

Quantification of nanoparticle-molecule interaction at a single-molecule level remains a daunting challenge, mainly due to ultra-weak emission from single molecules and the perturbation of the local environment. Here we report the rational design of an intraparticle-surface energy transfer (i-SET) process, analogous to high doping concentration-induced surface quenching effects, to realize single-molecule sensing by nanoparticle probes. This design, based on a $Tb^{3+}$-activator-rich core-shell upconversion nanoparticle, enables a much-improved spectral response to fluorescent molecules at single-molecule levels through enhanced non-radiative energy transfer with a rate over an order of magnitude faster than conventional counterparts. We demonstrate a quantitative analysis of spectral changes of one to four fluorophores tethered on a single nanoparticle through i-SET spectroscopy. Our results provide opportunities to identify photoreaction kinetics at single-molecule levels and provide direct information for understanding behaviors of individual molecules with unprecedented sensitivity.

[1] Institute for Composites Science Innovation, School of Materials Science and Engineering, Zhejiang University, Hangzhou 310027, China. [2] Division of Physics and Applied Physics, School of Physical and Mathematical Sciences, Nanyang Technological University, Singapore 637371, Singapore. [3] State Key Laboratory of Modern Optical Instrumentation, College of Optical Science and Engineering, Zhejiang University, Hangzhou 310027, China. [4] Key Laboratory for Organic Electronics and Information Displays, Institute of Advanced Materials, Nanjing University of Posts & Telecommunications, Nanjing 210023, China. ✉email: rdeng@zju.edu.cn

Characterization of molecular events using single-nanoparticle probes is of great interest as it can reveal molecular details inaccessible by ensemble observations[1]. For instance, single-particle experiments allow determination of surface modification heterogeneity on functional materials[2,3]. Single-nanoparticle probes can be used to monitor chemical and physical changes of individual molecules, especially for those occurring at the interface of catalytic materials[4,5]. Single-nanoparticle spectroscopy also provides an analytical tool that has the power to decode the complex biological processes for applications in bioimaging and biosensing[6–8].

Lanthanide-doped upconversion nanoparticles are capable of converting multiphoton near-infrared excitation into single-photon ultraviolet/visible emission[9,10]. These nanoparticles demonstrate significant advantages over other luminescence probes including non-blinking, high photostability, and good brightness uniformity at a single particle level. These advantages render them ideally suitable for single-particle imaging and sensing[11–14]. In particular, lanthanide-doped upconversion nanoparticles are potentially applicable for single-particle-based Förster resonance energy transfer (FRET) by probing the targeted energy transfer from the nanoparticles to fluorescent dye acceptors[15]. The zero background from fluorescent acceptors upon anti-Stokes excitations can permit highly selective recognition and ultrasensitive quantification of single-molecule targets. Recent works have shown great promises of using upconversion nanoparticles for FRET-based sensing in ensemble measurements[16–19]. However, these measurements generally require amplification of the target signal by increasing the number of acceptors linked to a nanoparticle. Although considerable efforts have been devoted to improving the sensitivity and stability of FRET sensitization such as constructing confined core–shell–shell architecture and increasing the longevity of organic fluorophores[20,21], for quantitation of molecular targets at single-molecule levels fundamental challenges remain in developing a highly sensitive upconversion nanoparticle capable of determining one-to-one binding between the nanoparticle and the molecular target.

Here, we propose a design principle that can be used to drastically increase the sensitivity of upconversion nanoparticles to fluorescence molecules, by combining high contents of activators and a spatially confined core-shell nanostructure. We demonstrate that a high concentration of activators can facilitate non-radiative resonance energy transfer from upconversion nanoparticles to surface-decorated fluorescent molecules by adopting a i-SET process (Fig. 1). This process can significantly enhance fluorescence signals from the targeted molecules, so that sensitized emission from a single molecule can be spectroscopically resolved at a single-particle level. Nevertheless, for conventional $Er^{3+}$, $Tm^{3+}$, or $Ho^{3+}$ activators co-doped with $Yb^{3+}$ or $Nd^{3+}$ ions as sensitizers[22–24], a high content of activator and sensitizer combination would generally lead to low luminescence efficiencies due to the concentration quenching effect[25–27]. In view of recent advances on using sensitizer-free shell coating to block prime concentration quenching paths via energy migration to surface quenchers[28–31], we envisage that efficient i-SET can be achieved by utilizing activator shell layers disfavouring energy transfer to the lower lying quenching sites such as –OH and $-CH_n$ vibration modes (Supplementary Figs. 1, 2). We choose $Tb^{3+}$ for validation as it features a relatively large energy gap ($\Delta E_{min} \approx 15,000 \text{ cm}^{-1}$) between the lowest emitting ($^5D_4$) and the ground ($^7F_J$) manifolds of $Tb^{3+}$. As a result, $Tb^{3+}$ would have high resistance to the low-energy surface quenchers. Moreover, the excited $^5D_4$ state of $Tb^{3+}$, situated in the visible spectral region, has substantial spectral overlapping with the absorption of many organic fluorophores. Hence, it is

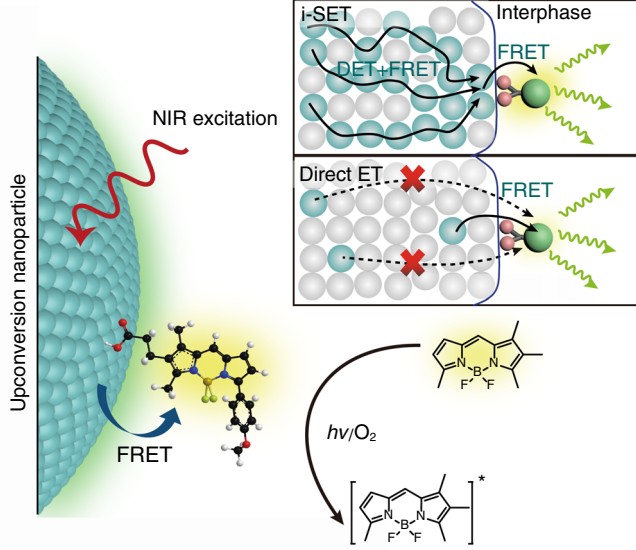

**Fig. 1 Schematic of single upconversion nanoparticle-based molecular sensing.** The sensing of individual molecules is realized by probing the FRET signal of nanoparticle-to-molecule energy transfer. The i-SET path shows energy transfer from nanoparticles with a high content of activators (represented as cyan spheres) in which excitation energy can fast migrate within the particles by either Dexter energy transfer (DET) or FRET and lead to promoted sensitization of adhesive molecules through intraparticle-to-surface FRET. The direct ET path indicates energy transfer from conventional upconversion nanoparticles having low doping concentration of activators in which activators buried inside the nanoparticle can hardly transfer their energy to the molecular acceptors due to the longer distance of separation.

possible to use $Tb^{3+}$ to tune the energy transfer pathways to leverage sensitization of organic fluorophores through single nanoparticles.

## Results

**Preparation of $Tb^{3+}$-based upconversion nanoparticles.** To validate our hypothesis, we synthesized $\beta$-NaYbF$_4$:Tb core nanoparticles containing 60 mol% of $Yb^{3+}$ sensitizers and 40 mol % of $Tb^{3+}$ activators, followed by coating with an activator-rich NaTbF$_4$ shell layer (Supplementary Fig. 3; see refs. [32,33]). The nanoparticles were confirmed to be a hexagonal phase by powder X-ray diffraction patterns (Supplementary Fig. 4). Transmission electron microscopy along with in situ energy-dispersive X-ray spectroscopy indicate high degree of monodispersity and well distinguishable core–shell configuration of the as-synthesized nanoparticles (Supplementary Fig. 5). To systematically investigate the impact of activator concentration on the energy transfer process, we also prepared a series of core-shell nanoparticles with $Tb^{3+}$ partially replaced by optically inert $Y^{3+}$ and $Lu^{3+}$ (Supplementary Fig. 6). Since $Y^{3+}$ and $Lu^{3+}$ do not have any energy state that matches emitting levels of $Tb^{3+}$ and $Yb^{3+}$ at the visible spectral region, they are expected to act as an inert matrix to prevent long-distance intraparticle energy transfer through the sublattice of the activators/sensitizers.

Under 980 nm excitation by a continuous-wave diode laser, the NaYbF$_4$:Tb@NaTbF$_4$ colloidal exhibits emissions at 490, 547, 590, and 620 nm, corresponding to $^5D_4 \rightarrow {}^7F_J$ ($J = 6, 5, 4, 3$) radiative transitions of $Tb^{3+}$, respectively (Supplementary Fig. 7). The emissions can be ascribed to a cooperative sensitization upconversion (CSU) process, in which energy from two excited $Yb^{3+}$ sensitizers simultaneously transfer to an adjacent $Tb^{3+}$

without the need of a lower lying intermediary state[34–36]. The presence of NaTbF$_4$ shell greatly enhances upconverted emission by a factor of two orders of magnitude, due to the block of nonradiative deactivation from Yb$^{3+}$ to surface anchored quenching sites by the shell protection (Supplementary Fig. 7). We then validated whether the high content of Tb$^{3+}$ brings concentration quenching by examining the lifetime changes of Tb$^{3+}$ as a function of dopant content. Since any of the quenching processes would bring additional energy depletion paths to Tb$^{3+}$ activators, the luminescence lifetime of Tb$^{3+}$ is supposed to decrease if concentration quenching occurs. As anticipated, we observed Tb$^{3+}$ lifetime at the $^5D_4$ state, measured by monitoring the luminescence decay at 547 nm, remains almost unaltered as the concentration of Tb$^{3+}$ in the shell layer increases (Supplementary Fig. 8). It supports the absence of surface quenching in these nanoparticles.

**Ensemble measurements of BDP-upconversion nanoparticle hybrids**. The core–shell nanoparticles were then decorated with carboxylic acid-functionalized borondipyrromethene dyes (BDP TMR carboxylic acid; denote as BDP for short) by adopting a two-step ligand exchange protocol (Fig. 2a and Supplementary Figs. 9–12). We used BDP for the study because of its high fluorescence quantum yield (0.95) and the large spectral overlapping between the BDP's peak absorption and the nanoparticle's maximum emission at 547 nm (Fig. 2b). The sensitization of the BDP dye by nanoparticles with different concentration of Tb$^{3+}$ was examined under excitation at 980 nm. By elevating Tb$^{3+}$ concentration (5–40 mol% in the core and 0–100 mol% in the shell), we observed a gradual increase in sensitized BDP emission at 573 nm and a concurrent decrease in Tb$^{3+}$ luminescence lifetime at 547 nm (Supplementary Figs. 13, 14). The results reveal a strong correlation between the BDP emission and the Tb$^{3+}$ concentration. Despite having similar particle sizes and emission profiles, we observed sensitized dye emission of NaYbF$_4$:Tb(40 mol%)@NaTbF$_4$ nanoparticles was 80 times stronger than that of NaYbF$_4$:Y,Tb(35,5 mol%)@NaLuF$_4$ nanoparticles, when both decorated with an average of ~34 BDP molecules per nanoparticle (Supplementary Fig. 15). The significantly amplified dye signal provides a much-improved detection limit to the molecule-nanoparticle interaction in the BDP-NaYbF$_4$:Tb(40 mol%)@NaTbF$_4$ conjugates. As shown in Fig. 2c, the sensitized emission of BDP can be clearly detected even when the loading concentration is as low as ~0.5 BDP per nanoparticle. The difference emission spectrum of nanoparticles with and without the BDP molecules (Fig. 2c inset) indicates that the sensitized BDP emission is almost identical to the downshifting emission of BDP through direct excitation. It confirms that the properties of the fluorophores are well retained at the nanoparticle's surface.

We attribute the enhancement of sensitized dye emission to intraparticle Tb$^{3+}$-Tb$^{3+}$ energy transfer, which promotes energy donating from the nanoparticles to the surface decorated BDP dyes. This is supported by a sharp decline in luminescence lifetime of Tb$^{3+}$ (from 1.93 to 0.44 ms) in NaYbF$_4$:Tb(40 mol%)@NaTbF$_4$ nanoparticles after addition of BDP (Supplementary Table 1). Besides, we also noted that at least 30 mol% of Tb$^{3+}$ in the shell is required to initiate efficient energy transfer from Tb$^{3+}$ to dye molecules (Supplementary Fig. 16 and Supplementary Table 2). It suggests a critical Tb$^{3+}$–Tb$^{3+}$ distance of ~6 Å for Tb$^{3+}$–Tb$^{3+}$ energy transfer in accordance with the reported critical distance for initiating energy migration between Tb$^{3+}$ ions[37,38]. To further verify the critical role of Tb$^{3+}$–Tb$^{3+}$ interaction for the i-SET process, we employed a semiempirical model, which is useful for predicting energy migration in

lanthanide-doped bulk phosphors[38,39], to describe the promoted dye sensitization by the nanoparticles. We modified the model to include a geometric distribution function of donor-acceptor pairs so that it can be used for heterogeneous finite systems like dye-nanoparticle hybrids (Supplementary Note 2; see ref. [40]). By assuming a Förster type of energy transfer from nanoparticles to BDP, we investigated the variation of energy transfer efficiency in the presence and absence of intraparticle energy transfer. For core–shell nanoparticles with a typical size of ~24 nm, the theoretical modeling shows the presence of i-SET in highly doped NaYbF$_4$:Tb(40 mol%)@NaTbF$_4$ nanoparticles increases energy transfer efficiency by a factor of 4–25 in comparison with NaYF$_4$:Yb,Er(18,2 mol%)@NaYF$_4$ control nanoparticles with only direct ET to the dye molecules, depending on the coverage (20-1 BDP) of dye molecules per nanoparticle (Fig. 2d). This is consistent with our experimental observations. For instance, we found that an average of 13 BDP on a NaYbF$_4$:Tb@NaTbF$_4$ nanoparticle (24 nm in diameter) can lead to about 74% efficiency (Supplementary Fig. 17 and Supplementary Tables 3, 4). By contrast, the measured energy transfer efficiency is only 16% for conventional NaYF$_4$:Yb,Er(18,2 mol%)@NaYF$_4$, the most representative low-doping nanoparticle widely applied for upconversion-based FRET studies[41,42], when decorated with a similar concentration of dye molecules (Fig. 2d and Supplementary Figs. 18, 19). Moreover, our investigation also confirms that the high activator content is a viable strategy to promote energy donation from upconversion nanoparticles of various particle sizes (Fig. 2d inset, Supplementary Figs. 20–22, Supplementary Table 5).

We next used the experiment results to refine the theoretical models to reveal the underlying physical processes. As illustrated in Fig. 2e, the i-SET involves stochastic energy migration through Tb$^{3+}$ sublattice and subsequently FRET from the $^5D_4$ state of a nearby Tb$^{3+}$ to the singlet excited state of a BDP molecule given an Förster critical distance of ~6 nm for the Tb$^{3+}$-BDP pairs. We can calculate a rate of $9 \times 10^7\,s^{-1}$ for the energy migration between nearest Tb$^{3+}$ neighbors and a rate of $3 \times 10^4\,s^{-1}$ for FRET from the nearest Tb$^{3+}$ to BDP (see Supplementary Note 2 for details). Given the intrinsic decay dynamics of ~500 s$^{-1}$, the energy transfer rates kinetically outcompete other decay pathways, leading to reasonably efficient dye sensitization by highly-doped nanoparticles in comparison with the conventional direct FRET model (Supplementary Fig. 23). In addition, the model calculations also predict that BDP molecules would compete for accepting energy from the i-SET nanoparticle donors, as each additional molecule will reduce the probability of energy transfer to existing molecules. As shown in Fig. 2f, the emission intensity of each dye molecule decreases as the dye coverage increases. The single dye contribution was found to drop by 60% when the coverage of dye molecules increased from 1 to 7 molecules per nanoparticle. This behaviour is different to the direct FRET in which energy transfer to a dye acceptor is independent to the addition of other molecules in the same concentration range. The results imply that the highly-doped nanoparticles will be more sensitive to small molecules at a low concentration region, which would be particularly suitable for single molecule detection.

**i-SET spectroscopy for single BDP characterization**. The experimental results along with the calculations encourage us to demonstrate further the viability of identifying molecule-nanoparticle interaction through i-SET-based single-particle spectroscopy. As a proof-of-concept, BDP-modified upconversion nanoparticles (2.5 pM) in cyclohexane were drop-casted onto a glass substrate and subsequently imaged by a homemade confocal scanning microscopic system (Fig. 3a and Supplementary

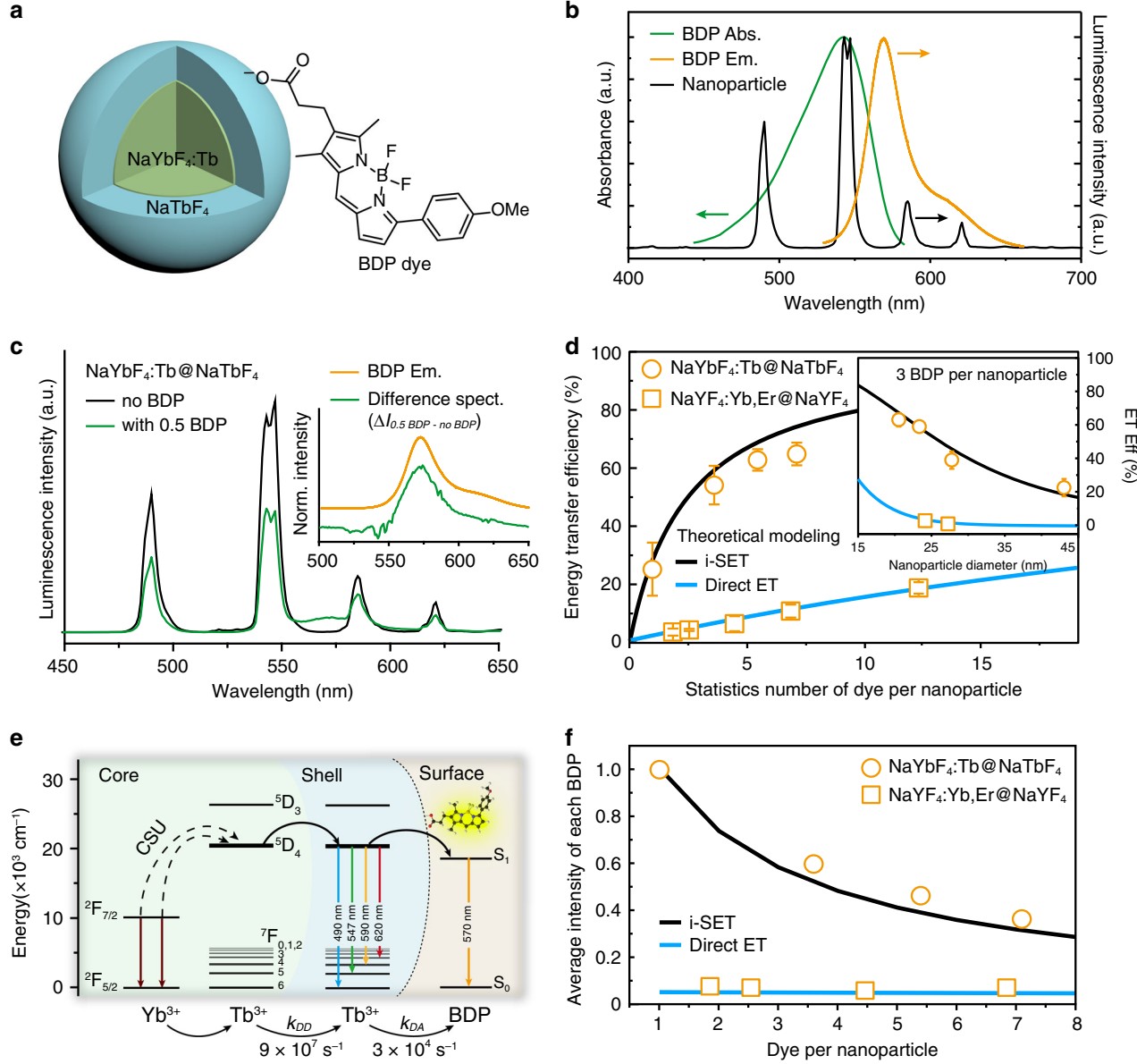

**Fig. 2 Experimental and theoretical investigations of energy transfer between upconversion nanoparticles and organic dyes. a** Schematic illustration of a NaYbF$_4$:Tb(40 mol%)@NaTbF$_4$ core-shell upconversion nanoparticle coupled with a carboxylic acid-functionalized borondipyrromethene dye (BDP dye) in our study. **b** Normalized UV–vis absorption (Abs.) and emission (Em.) spectra of BDP dyes, and upconversion luminescence (PL) spectrum of NaYbF$_4$:Tb(40 mol%)@NaTbF$_4$ nanoparticles. **c** Upconversion luminescence spectra of NaYbF$_4$:Tb(40 mol%)@NaTbF$_4$ nanoparticles conjugating with and without BDP conjugation. The inserted diagram shows shows a comparison between the direct emission spectrum of free BDP molecules at 365 nm excitation and the sensitized emission spectrum of coupled BDP obtained by calculating the spectral differences between emission spectra of the nanoparticles with and without 0.5 BDP conjugation. **d** Theoretical plots and experimental data of energy transfer efficiencies for NaYbF$_4$:Tb(40 mol%) @NaTbF$_4$ and NaYF$_4$:Yb,Er(18,2 mol%)@NaYF$_4$ core–shell nanoparticles with the same diameter of 24 nm as a function of average BDP molecules per nanoparticle. Inset is energy transfer (ET) efficiency plotted as a function of particle size for nanoparticles decorated with an average of three BDP per particle. The error bars represent one standard deviation of three parallel experiments. **e** Proposed energy transfer mechanism describing the i-SET from NaYbF$_4$:40%Tb@NaTbF$_4$ core–shell nanoparticles to BDP dyes. **f** Simulated and experimental average luminescence intensity of single BDP molecule as a function of BDP coverage on individual nanoparticles.

Fig. 24), following a reported protocol for imaging single upconversion nanoparticles[12]. Under 980 nm excitation by a focused laser beam at $10^5$ W cm$^{-2}$ (Supplementary Fig. 25), we obtained microscope images of multiple bright spots with sizes close to the diffraction-limit (Fig. 3b). Most of the spots are observed to have a uniform and stable brightness with an average detectable signal of around 1000 counts per second (Supplementary Figs. 26, 27). We confirmed that these spots are correspondent to emission from single nanoparticles except for a small

fraction of dimers and other aggregates, as evidenced by the electron microscope images (Supplementary Figs. 28, 29). We then examined the emission spectra of each spots to gain information from individual nanoparticles. As anticipated, the sensitized dye emission can be clearly visualized from BDP-decorated NaYbF$_4$:Tb@NaTbF$_4$ nanoparticles (Fig. 3b). Notably, despite having similar total brightness, the luminescence spectra were found to be heterogeneous with clear particle-to-particle variations in peak intensity of BDP emission at 573 nm (Fig. 3c,

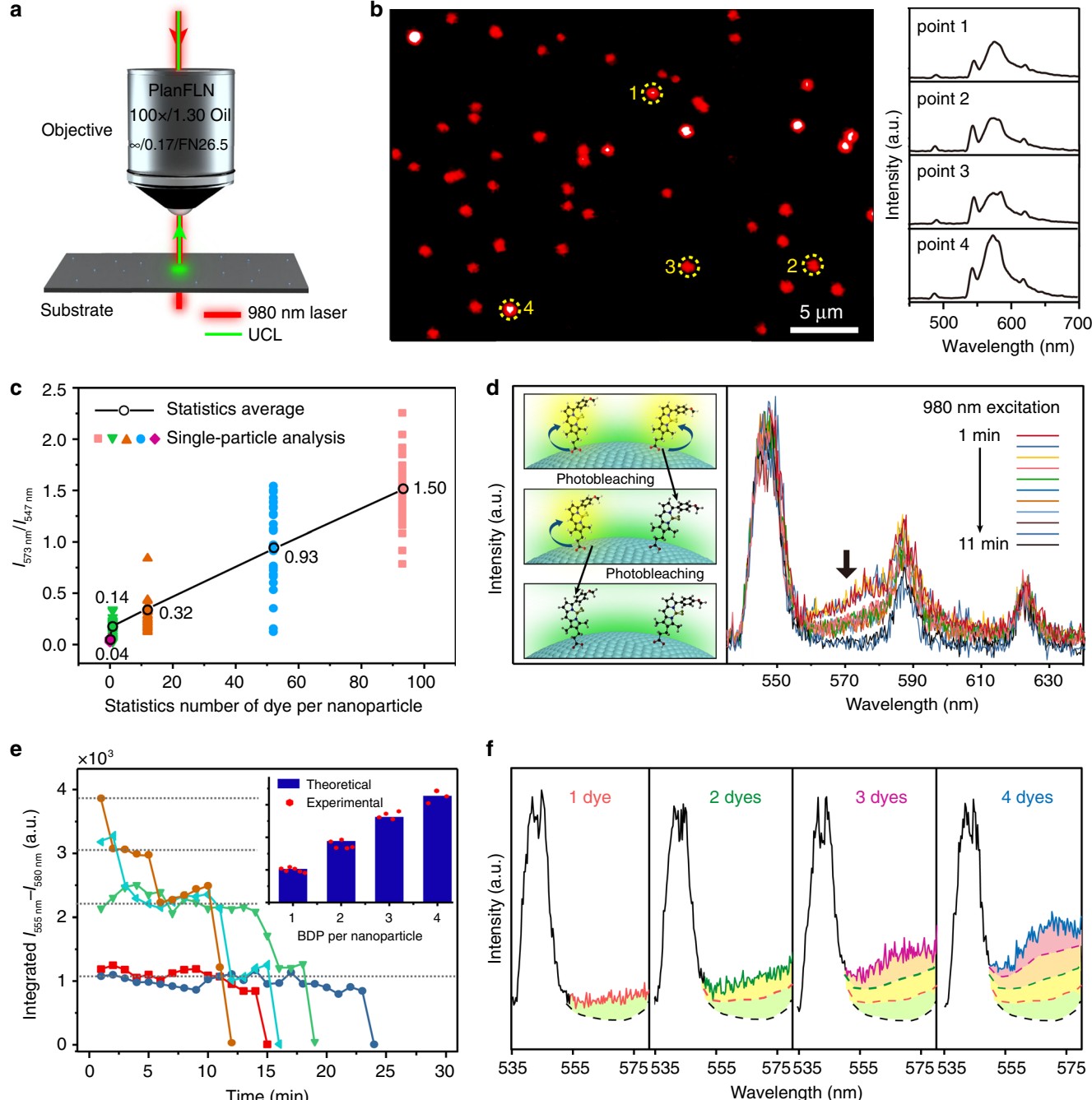

**Fig. 3 Single-particle characterization of BDP dye-modified upconversion nanoparticles through i-SET spectroscopy. a** Schematic of the experimental design for detecting single-upconversion luminescence (UCL) from of dye-decorated upconversion nanoparticles using a confocal microscope imaging system. **b** A typical confocal microscope image of $NaYbF_4$:Tb(40 mol%)@$NaTbF_4$ nanoparticles loaded with an average of ~92 BDP molecules per nanoparticle (left panel), and corresponding single-particle upconversion luminescence spectra (right panel) recorded at the points marked with yellow dashed circles. **c** Plots of sensitized BDP emission to $Tb^{3+}$ emission intensity ratios measured at single-particle level from a series batches of BDP-decorated $NaYbF_4$:Tb(40 mol%)@$NaTbF_4$ nanoparticles with different average coverage ratios of dye molecules. Each of the colour dots represent an individual spectral measurement from a randomly picked nanoparticle, and the black circles represent the average ratios of individual measurements from the same sample glass substrates. **d** Time-dependent upconversion luminescence spectra of a typical BDP-decorated upconversion nanoparticle showing discrete two-steps photobleaching of sensitized BDP emission at around 573 nm under 980 nm irradiation. The inserted scheme illustrates the step-wise single-molecule photobleaching of two BDP molecules at the surface of a nanoparticle. **e** Time-dependent integrated emission intensity changes of BDP recorded from several individual single-particle measurements. The inset shows a comparison between the theoretically calculated dye emission intensity ratios and the experimental intensity ratios extracted from the single-particle measurements. **f** Single-particle upconversion luminescence spectra of individual $NaYbF_4$:Tb(40 mol%)@$NaTbF_4$ nanoparticles conjugated with 1, 2, 3, and 4 active dye molecules, respectively. The nanoparticle-sensitized BDP emissions are highlighted in colours.

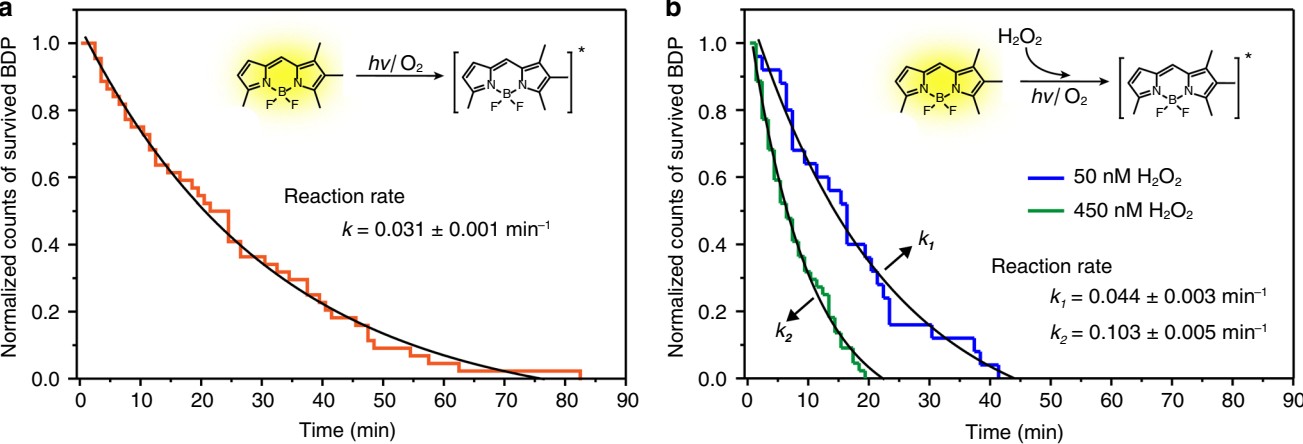

**Fig. 4 Kinetic investigation of single-molecule photoreactions. a** Normalized counting of survived BDP molecule collected from 45 individual single-molecule measurements as a function of irradiation time. It reveals photobleaching kinetics of the single-molecule reaction of BDP at the surface of $NaYbF_4$:Tb(40 mol%)@$NaTbF_4$ upconversion nanoparticles. **b** Plots of normalized counting of survived BDP molecule versus irradiation time indicating the single-molecule reaction of BDP accelerates after exposure to a different amount of $H_2O_2$.

Supplementary Figs. 30–34). This observation suggests the changes in dye coverage on each nanoparticle, which provides direct evidence for the heterogeneity of surface modification to the nanoparticles. On the other hand, even with a relatively high BDP coverage (an average of 95 per nanoparticle) the control sample of BDP-decorated $NaYF_4$:Yb,Er(18,2 mol%)@$NaYF_4$ nanoparticles do not show any detectable dye emission signal in single-particle measurements (Supplementary Figs. 35, 36). One explanation for the absence of sensitized emission would be the much slower energy transfer rate from the conventional Yb/Er co-doped nanoparticles to BDP. The observation again supports the necessity of our material design for extending the detection limit of single-particle FRET.

We then investigated the spectral changes of individual $NaYbF_4$:Tb@$NaTbF_4$ nanoparticles conjugated with only few BDP molecules (an average of ~0.9 BDP per nanoparticle in stock solutions) as a function of laser exposure time. Interestingly, while the luminescence of $Tb^{3+}$ from the nanoparticles is stable for hours without noticeable changes, the sensitized emission from the BDP dyes was found to decrease in discrete steps (Fig. 3d). This is very different from ensemble measurements in BDP-decorated nanoparticle solution, where a continuous decrease in DBP emission was observed. By examining 31 individual nanoparticles, we identified 5, 4, 5, and 2 nanoparticles with distinct 1-step, 2-steps, 3-steps, and 4-steps BDP quenching, respectively, along with 15 nanoparticles having no sensitized dye emission (Fig. 3e and Supplementary Figs. 37, 38). The nanoparticles show consistent intensity at the BDP emission region when they are at the same quenching levels. The experiment intensity ratios for each quenching step fit well with the simulated intensity ratios of sensitized molecular emission by our model calculation (Fig. 3e inset). These results strongly support that we have observed single-molecule behaviors occurring at the interface of the nanoparticles. Herein, each step of spectral change should indicate the occurrence of a photobleaching reaction for a certain BDP molecule[43]. Given the high contrast capability of measuring single-molecule activities, this approach allows us to spectroscopically discriminate the exact loading number of dye molecules on individual upconversion nanoparticles (Fig. 3f). Importantly, unlike the digital relationship of conventional single-molecule measurements[1], the sensitized BDP emission does not follow integer-times-relation to the number of dyes. Compared with the emission intensity of one-to-

one bonded BDP-nanoparticle hybrids, the integrated intensities of BDP were observed to increase by only 1.9, 2.6, and 3.3 times when 2, 3, and 4 molecules were bonded to a nanoparticle, respectively. This is reasonable because the addition of another molecule would bring competition to FRET from the nanoparticle to each molecular acceptors. As a result, the emission probability of each dye decreases as the number of dye molecules increases.

In addition, the discrete quenching steps can be used to further characterize the reaction properties of isolated BDP molecules on the nanoparticles. For individual measurements, the luminescence quenching of BDP looks like entirely random events, as we observed some of the molecules underwent photobleaching within few minutes while others kept emitting for up to more than an hour without noticeable spectral change. Nonetheless, the statistical results collected from a large number of measurements reveal that the reactions of individual BDP still obey a concentration-dependent reaction kinetics. By plotting the counts of survived BDP molecules against laser exposure time followed an exponential decay function, a first-order reaction kinetics of the single-molecule photobleaching of BDP is derived with a reaction rate constant of $0.031 \pm 0.001\ min^{-1}$ (Fig. 4a and Supplementary Fig. 39). Furthermore, we found that the decay of BDP became faster when we added a small amount of $H_2O_2$, an oxidant supposed to increase the probability of photobleaching, to the BDP-nanoparticle stock solution before dropcasting the samples. As indicated in Fig. 4b, we obtained reaction rates of $0.044 \pm 0.003$ and $0.103 \pm 0.005\ min^{-1}$ for samples added with 50 and 450 nM $H_2O_2$, respectively. The results suggest the photobleaching of BDP is very sensitive to local environmental changes.

## Discussion

In conclusion, we report the generation of efficient single-molecule upconversion nanoprobes by applying a high-activator-content-promoted i-SET approach. We demonstrate that factors known to be harmful to luminescence efficiency in accelerating surface quenching may be beneficial to energy donation from the nanoparticles to their surface-modified molecular acceptors. Our results establish a methodology to enable precise control over the energy transfer pathways between luminescence nanoparticles and various types of functional targets. The significant enhanced nanoparticle-to-molecule interaction enables us to quantitatively probe single-molecule behaviors on a nanoparticle with ultimate

precision. These nanoprobes should be potentially applicable to the evaluation of the effects of different surface modifications on the heterogeneity of organic-nanoparticle hybrids. They may also be useful for understanding photochemical properties of individual molecules on nanomaterial surfaces. Our findings could open new avenues for the design of high-performance single-particle upconversion spectroscopy, particularly suitable for applications in photosensitization, photocatalysis, and single-molecule characterization.

## Methods

**Preparation of BDP dye-modified upconversion nanocrystals**. The $NaLnF_4$ (Ln = Y, Tb, Yb, Lu) core nanoparticles were synthesized by a well-established co-precipitation method. The oleate-capped core–shell nanoparticles were subsequently synthesized by a layer-by-layer hot injection method using the as-synthesized core nanoparticles as seeds and cubic phase $NaLnF_4$ nanocrystals as shell precursors. Further experimental details are available in the Supplementary Methods. For the preparation of BDP-decorated nanoparticles, 0.5 mL cyclohexane solution of oleate-capped core-shell nanoparticles (1.1 μM) was mixed with 2.5 mL THF solution containing BDP (1–40 μM) in a 25 mL flask. The mixture was heated to reflux at 50 °C for 2 h with vigorous stirring under nitrogen protection. The conjugates were collected by centrifugation (20,000 rpm, 20 min), washed with ethanol twice and re-dispersed in cyclohexane (0.7 μM, 2.5 mL). The dye loading concentration can be controlled by adjusting the concentration of BDP in the added solution.

**Sample preparation for single-nanoparticle microscopic imaging**. In a typical procedure, the nanoparticle solution in cyclohexane was diluted to 2.5 pM. The diluted solution (15 μL) was then dropcasted onto a clean cover-glass (2 cm × 2 cm), and carefully rinsed using 20 μL cyclohexane. After the evaporation of cyclohexane, the as-prepared samples were imaged immediately by using a confocal microscope.

**Single-nanoparticle microscopy**. The upconverted luminescence from individual nanoparticles was characterized by a homemade confocal scanning optical microscope (Supplementary Fig. 24). A beam of 980 nm laser from a single-mode CW diode laser was tightly focused on the sample through a 100× oil objective (NA 1.35). The upconversion luminescence from the sample was collected through the same objective and the 980 nm excitation was filtered out with a short pass filter (805 nm cutoff). The emission signal was detected either by a Single Photon Counting Module (EXCELITAS, SPCM-AQRH-14-FC34229) or by a spectrometer (Princeton Instruments, ProEM) equipped with a CCD camera (eXcelon3).

## Data availability
All relevant data that support the findings of this work are available from the corresponding author upon reasonable request.

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

## Acknowledgements

The work is supported by National Key Research and Development Program of China (2018YFB0703803), Zhejiang Provincial National Natural Science Foundation of China (LR19B010002), National Natural Science Foundation of China (21801222 and 51872256), and the open research fund of Key Laboratory for Organic Electronics and Information Displays. We thank Prof. Xiaogang Liu for valuable suggestions on the study.

## Author contributions

J.Z. and R.D. conceived the project and wrote the manuscript. R.D., W.G. and J.Q. supervised the project and let the collaboration efforts. J.Z. carried out the material synthesis and ensemble spectral measurements. L.C. performed the model calculations. J.Z., L.D., and Z.M. conducted the single-particle experiments with input from X.L., W.G., and J.Q. All authors discussed the results and commented on the manuscript.

## Competing interests

The authors declare no competing interests.
