## [Peer Review File · Nature Communications]

REVIEWER COMMENTS

Reviewer #1 (Remarks to the Author):

This manuscript addresses experimental realization of highly efficient Forster resonant energy transfer from nanoparticles to molecules at the surface by doping Tb³⁺ inside the nanocrystals/nanoparticles, as well as resultant great enhancement of photoluminescence from the surface molecules. Tb has advantages compared to other lanthanide elements because vibrational levels of the surface molecules are well isolated from the electronic levels of the Tb³⁺ (that are relevant to the excitations by NIR) and the electronic levels of the surface molecules, which prevents fast non-radiative decay. The authors unambiguously present their key results through many different types of measurements and control experiments by varying the dopant types, Tb concentration, the number of dye molecules, the number of BDP molecules (surface molecules), and local environment. In addition, a phenomenological model adapted from Ref.36 supports the experimental data with several fitting parameters. The manuscript is very well written and it is recommended for publication with the following minor revision.

There are two typos:

- 1) On page 3, right hand side, 2nd line: 4D5  5D4 in "From the 5D4 state of a nearby Tb³⁺...."
- 2) In Ref.36: The page number is incorrect.

There are a few scientific questions on the manuscript.

- 1) On page 3, the authors mentioned that "the calculation shows the presence of intraparticle Tb³⁺-Tb³⁺ interaction increases energy transfer efficiency by a factor of 4-25, depending on the coverage (20-1 BDP per nanoparticle) of dye molecules (Fig. 2d)". But if this is from a simple model with fitting parameters that the authors presented in the supporting information, it would be good that the authors explain a bit more about this statement. Experimentally it is not possible to measure the distance between the neighboring Tb³⁺ dopants inside the nanoparticles. Perhaps the authors have drawn this conclusion from the concentration of the Tb³⁺ for a given nanoparticle size.
- 2) Interestingly, in Fig.S5, the peak from 5D4 to 7F5 is higher than that from 5D4 to 7F6, although 7F6 is the ground electronic level of Tb³⁺. Is there any chemical or physical reason for this?
- 3) From the experimental data, it is clear that the energy is transferred from the Tb³⁺ to the surface molecules. However, how would the authors know that the excited energy is transferred from one Tb³⁺ to nearest neighboring Tb³⁺ site? Isn't just a higher concentration of Tb³⁺ enough to explain the experimental data?

Reviewer #2 (Remarks to the Author):

The article "Single-molecule photoreaction quantitation through intraparticle-surface energy transfer (i-SET) spectroscopy" by J.Zhou et al. presents intraparticle-surface energy transfer using Tb³⁺ ions aiming to study single fluorophores tethered on a single NP, in opposite to existing studies, where number of acceptor fluorophores is increased. The idea of removing Yb³⁺ ions (highly susceptible to surface quenching by overtones of O-H vibrations) and exploiting high energy gap Tb³⁺ as donor) is smart, original and novel and it is convincingly and nicely presented. The article is very well written/prepared, experiments are convincing and sufficiently well planned to demonstrate the hypotheses. However some points require correction and more in depth explanations.

1. What is the power density of CW 980 nm excitation in Fig.S5? How does it compare to excitation intensities required for conventional YbEr UC ? The question arises from a fact, the CSU is typically 100-times weaker than e.g. YbEr UC. In the same time, stronger pumping at 980 nm (required for CSU) could lead to increased and undesired heating in water environments - did authors

measure/notice such effects ?

2. why 5%Tb@... sample (Fig.S11) shows double exponential behavior while the samples with higher Tb content do not?
3. Why infinitely small YbEr@Y NPs show FRET efficiency, which is dependent on number of BDP acceptors, unlike similarly small YbTb@Tb NPs ? I cannot see a reason for that, which somehow questions the modelling results in its current form.
4. The history of Yb-Tb upconversion is slightly longer and rich.
5. Fig.2f - it is not clear and not explained why such behavior is observed and what it really means. More in depth discussion is needed, including the discussion about the radiative rates of A and D
6. Fig.4 - what are the expected reasons for >60 minutes photo stability when most A molecules photo bleach faster. I know this is statistics, but can the authors draw deeper conclusions about this interesting observation. Does this behavior change (and how) with excitation intensity, number of A molecules per NP? What is the environment the measurements were made in ? Most biosensors require water based conditions.
7. I found numerous figures and tables in SI, which are not commented in main article (which would be desirable, so don't skip these figures, but add comments). For example - how the number of A dyes per NP has been calculated ? Some more detailed information about statistics of number of A molecules per NPs should be provided and discussed.
8. The authors presented acceptor molecules directly anchored to the surface of NP. But actual biosensors require bio-specific molecules such as antibody-antigen or DNA molecules, which will increase the distance between superficial Tb donors to the acceptor present a few nm away (due to intermediate bio-recognition molecules). What is the Forster distance for the given Tb-BDP molecules ? Did authors test any distance dependence for this interesting NPs ?

Some language mistakes have been found and need correction (e.g. "intensity of single BDP molecule_S_", "reaches _to_ a maximum" and others).

Overall, I liked the concept and the quality of the presented results. I find it novel and I am sure this will be inspiration for other scientists working in the field of bio sensing and lanthanide doped applications.

Reviewer #3 (Remarks to the Author):

This study by Deng and coworkers demonstrates single-particle energy transfer between lanthanides in upconversion nanoparticles (UCNPs) and surface-bound BODIPY fluorophores, optimizing Ln content and demonstrating wire-like Ln chains that mediate energy transfer from the core of the nanoparticle to surface fluorophores. This is an interesting and dense study (with about 40 SI figures), and it includes some demanding single particle experiments. Overall, I would say the characterization of these nanoparticles is rigorous and interesting, particularly the impressive experiments in which the fluorophores are bleached one at a time to reveal stoichiometry. Some of the design work of how to optimize signal from the fluorophores is similar to previous studies from the past few years, and my sense is the paper should emphasize its key strength, which is the single particle work. I think a manuscript with some revisions would make a compelling addition to the UCNP and single particle imaging literature.

Specific comments:

1) Much of the nanoparticle design work here has been explored in detail before, including similar shell structures of NaLnF₄ (JACS 141, 42 (2019)) and optimization of energy transfer to fluorescent reporters by varying Ln doping and surface dye attachment (Opt Mat 84, 345 (2018)). The latter paper also has data on enhanced fluorophore photostability, which is somewhat at odds with the current study. It would be useful to put the current findings in context of this prior work.

2) "As shown in Figure 2c, even at a low BDP concentration of ~0.5 BDP per nanoparticle, the sensitized emission from BDP molecules is still detectable with a signal-to-noise ratio of up to 30."

Signal:noise is not the correct term for the calculation in 2c. For cuvette measurements, the noise of a fully background-subtracted spectrum is 0, so signal:noise would be infinite. More appropriate here would be a difference spectrum to see if the peak matches the free BODIPY dye emission, and to calculate the ET efficiency.

This BODIPY has an unfortunately small Stokes shift, which may make calculations of ET efficiency difficult.

3) The discussion of i-SET (a new term?) and cartoons in Fig 1 leave the impression of direct-contact energy transfer, such as Dexter e- transfer. I don't think this is likely, nor is there evidence for this. Even if an i-SET Ln wire ET process moves energy from the nanocrystal interior toward the surface, ET between Ln and fluorophore is still most likely a FRET process. The authors might consider clearer ways to distinguish FRET from the core vs FRET from the shell. If there is evidence of non-FRET processes such as Dexter, the authors should highlight it.

4) Given the direct contact between BODIPY and heavy atom shell, as well as rapid O₂-mediated quenching, do the authors believe triplets are forming in the BODIPY? There is evidence of this in NIR dye sensitization of Yb/Er UCNPs, although this remains somewhat controversial.

5) "Nevertheless, for conventional Er³⁺, Tm³⁺, or Ho³⁺ activators co-doped with Yb³⁺ or Nd³⁺ ions as sensitizers²¹⁻²³, a high content of activator and sensitizer combination would generally lead to low luminescence efficiencies of the nanoparticles due to concentration quenching effects²⁴."

There is a growing literature at this point on high-Ln UCNPs, and it has become clear that concentration quenching is not a significant factor in core/shell UCNPs. The authors seem to recognize that adding Yb-free shells necessarily cuts off the prime energy loss pathway. See e.g., JACS 139, 3275 (2017) and Nat Comm 9, 3082 (2018) for previous, in-depth discussions of high Ln core/shell UCNPs and why the concept of concentration quenching is dated. I think the discussion in the current paper can take this prior work into account to substantially shorten this part of the manuscript.

Reviewer #1:

This manuscript addresses experimental realization of highly efficient Forster resonant energy transfer from nanoparticles to molecules at the surface by doping Tb^{3+} inside the nanocrystals/nanoparticles, as well as resultant great enhancement of photoluminescence from the surface molecules. Tb has advantages compared to other lanthanide elements because vibrational levels of the surface molecules are well isolated from the electronic levels of the Tb^{3+} (that are relevant to the excitations by NIR) and the electronic levels of the surface molecules, which prevents fast non-radiative decay. The authors unambiguously present their key results through many different types of measurements and control experiments by varying the dopant types, Tb concentration, the number of dye molecules, the number of BDP molecules (surface molecules), and local environment. In addition, a phenomenological model adapted from Ref.36 supports the experimental data with several fitting parameters. The manuscript is very well written and it is recommended for publication with the following minor revision.

Response: We thank Reviewer 1 for the positive comments to our work.

There are two typos:

- 1) On page 3, right hand side, 2nd line: $^4D_5 \rightarrow ^5D_4$ in “From the 5D_4 state of a nearby Tb^{3+}”
- 2) In Ref.36: The page number is incorrect.

Response: We thank the reviewer for spotting these mistakes. We have made correction in the revised manuscript.

There are a few scientific questions on the manuscript.

- 1) On page 3, the authors mentioned that “the calculation shows the presence of intraparticle Tb^{3+} - Tb^{3+} interaction increases energy transfer efficiency by a factor of 4-25, depending on the coverage (20-1 BDP per nanoparticle) of dye molecules (Fig. 2d)”. But if this is from a simple model with fitting parameters that the authors presented in the supporting information, it would be good that the authors explain a bit more about this statement. Experimentally it is not possible to measure the distance between the neighboring Tb^{3+} dopants inside the nanoparticles. Perhaps the authors have drawn this conclusion from the concentration of the Tb^{3+} for a given nanoparticle size.

Response: Yes, this conclusion is obtained from model calculations with fitting parameters. As noted by the reviewer, we compared the calculated energy transfer efficiencies for nanoparticles with a given particle size of 24 nm and different concentration of activators. As shown in Fig. 2d, the modelling results indicate that the nanoparticle with highest activator concentration (which should have strongest Tb^{3+} - Tb^{3+} interaction) has a higher energy transfer efficiency than that with lowest activator concentration (a simulated nanoparticle without intraparticle energy transfer). Note that in Fig. 2d we also performed experimental comparison for nanoparticles with $(NaYbF_4:Tb@NaTbF_4)$ NPs) and without $(NaYF_4:Yb,Er@NaYF_4)$ NPs) high content of activators. The experimental data follows the same trends to the simulated curves, confirming the reliability of our model calculations.

For clarity, we have revised the corresponded statement with more experimental details:

“For core-shell nanoparticles with a typical size of ~24 nm, the theoretical modeling shows the presence of i-SET in highly doped $NaYbF_4:Tb(40\text{ mol}\%)@NaTbF_4$ nanoparticles increases energy transfer efficiency by

a factor of 4-25 in comparison with NaYF₄:Yb,Er(18,2 mol%)/NaYF₄ control nanoparticles with only direct ET to the dye molecules, depending on the coverage (20-1 BDP) of dye molecules per nanoparticle (Fig. 2d).”

2) Interestingly, in Fig.S5, the peak from ⁵D₄ to ⁷F₅ is higher than that from ⁵D₄ to ⁷F₆, although ⁷F₆ is the ground electronic level of Tb³⁺. Is there any chemical or physical reason for this?

Response: We appreciate the careful reading by the reviewer. The higher emission intensity for the ⁵D₄ to ⁷F₅ transition is due to the electric dipolar selection rules. It is dependent on multiple effects such as crystal field splitting, radiative transition probability, branching ratio, and peak emission cross-section of different transitions. Indeed, it is generally accepted that the ⁵D₄ to ⁷F₅ transition is the dominated radiative deactivation process of the ⁵D₄ state in many Tb³⁺-doped luminophores. Some previous studies have particularly demonstrated the similar fluorescence and radiative properties of Tb³⁺, providing explicit explanation about what the reviewer questioned (see e.g. *Opt. Mater.*, 2008, **30**, 1343–1348; *Opt. Commun.*, 2009, **282**, 3028–3031; *J. Mater. Chem.*, 2012, **22**, 22989–22997).

3) From the experimental data, it is clear that the energy is transferred from the Tb³⁺ to the surface molecules. However, how would the authors know that the excited energy is transferred from one Tb³⁺ to nearest neighboring Tb³⁺ site? Isn't just a higher concentration of Tb³⁺ enough to explain the experimental data?

Response: We believe the energy transfer among nearest Tb³⁺ neighbors is a necessary process for promoting FRET from the nanoparticles to the molecules. To clarify this point further, we would like to bring the reviewer's attention to the Figure S11e-o and Figure S12b in Supplementary Information. For NaYbF₄:Tb(40 mol%)/NaLuF₄:Tb (X mol%; X=0-100) core-shell nanoparticles, we observed a continuous increase in sensitized emission intensity of BDP in accordance with a concurrent decrease in luminescence lifetime of Tb³⁺, indicating the FRET efficiency of the nanoparticle kept increasing when the Tb³⁺ concentration in the shell layer increases from 0 to 100 mol%. Since there is no Yb³⁺ sensitizers in the shell layer of these nanoparticles, the only possible way to activate Tb³⁺ ions in the shell must be energy transfer from Tb³⁺ from the core to the shell. This result unambiguously suggests that Tb³⁺-Tb³⁺ energy transfer is a necessary step to migrate excitation energy to the shell layer for boosting energy transfer from a nanoparticle to surface anchored molecules.

Reviewer #2:

The article "Single-molecule photoreaction quantitation through intraparticle-surface energy transfer (i-SET) spectroscopy" by J.Zhou et al. presents intraparticle-surface energy transfer using Tb³⁺ ions aiming to study single fluorophores tethered on a single NP, in opposite to existing studies, where number of acceptor fluorophores is increased. The idea of removing Yb³⁺ ions (highly susceptible to surface quenching by overtones of O-H vibrations) and exploiting high energy gap Tb³⁺ as donor is smart, original and novel and it is convincingly and nicely presented. The article is very well written/prepared, experiments are convincing and sufficiently well planned to demonstrate the hypotheses. However some points require correction and more in depth explanations.

Response: We are grateful with the reviewer's recognition and positive comments of our work.

1. What is the power density of CW 980 nm excitation in Fig.S5? How does it compare to excitation

intensities required for conventional YbEr UC? The question arises from a fact, the CSU is typically 100-times weaker than e.g. YbEr UC. In the same time, stronger pumping at 980 nm (required for CSU) could lead to increased and undesired heating in water environments - did authors measure/notice such effects?

Response: We thank the reviewer for the critical comment. Indeed, we did not use a super high excitation power in the study. The power density of the CW 980 nm excitation is around 100 W/cm² in Fig. S5, which is moderate and should be suitable for measurements in aqueous environments. Although, the UC emission intensity of the YbTb nanoparticles was found to be around 20 times weaker (in the strongest peak intensity) than the most intense YbEr core-shell UC nanoparticles (see Figure R1 below), the luminescence of the current YbTb based system can still be easily recorded and analyzed by our spectroscope (with $\sim 8 \times 10^4$ counts of signal versus ~ 50 counts of background noise) under excitation of 100 W/cm². For clarity, we have added the excitation condition in the figure caption in Fig. S5.

Figure R1. Upconversion luminescence spectra of NaYF₄:Yb,Er(18,2 mol%)/@NaYF₄ and NaYbF₄:Tb(40 mol%)/@NaTbF₄ core-shell nanoparticles under 980 nm excitation at 100 W/cm² power density. Inset: the digital luminescence photographs showing corresponding nanoparticle colloidal.

2. Why 5%Tb@... sample (Fig.S11) shows double exponential behavior while the samples with higher Tb content do not?

Response: The reviewer has a very careful reading to our results. After a careful evaluation, we find that the double exponential decay for the 5%Tb@... sample (Figure R2) is more likely due to the emission from a trace impurity, such as Er³⁺. As shown in Figure R3, the photoluminescence spectrum of the 5%Tb@... sample contains impurity peaks at 475 and 521 nm which should be correspondent to emission from Tm³⁺ and Er³⁺, respectively. This is probably because energy transfer from Yb³⁺ to Tb³⁺ is insufficient at low Tb³⁺ concentration, so that ETU from excited Yb³⁺ to the trace amount of impurities become possibly efficient. We feel that the impurity effect is almost inevitable for sample with low doping concentration of activators, as we can see clear ETU emission peaks of Er³⁺ and Tm³⁺ even for the pure NaYbF₄ sample without any intentionally doped activators (Figure R3a). Nevertheless, the impurity emission significantly decreases after increasing the concentration of Tb³⁺ to more than 10 mol%, so it would not affect the main conclusion

of our study. For clarity, we have added a short interpretation to the double exponential decay behavior of the 5%Tb@ sample indicating it is from emission of impurities in the figure caption in Fig.S11.

Figure R2. 547 nm luminescence decay curve of NaYF₄:Yb,Tb(60,5 mol%)@NaLuF₄ nanoparticles.

Figure R3. The upconversion spectrum of **a**, NaYbF₄ and **b**, NaYF₄:Yb,Tb(60,5 mol%)@NaLuF₄ nanoparticles under 980 nm excitation.

3. Why infinitely small YbEr@Y NPs show FRET efficiency, which is dependent on number of BDP acceptors, unlike similarly small YbTb@Tb NPs? I cannot see a reason for that, which somehow questions the modelling results in its current form.

Response: This is because different assumptions were made for the two models. For the direct FRET model, we assume that the distribution of acceptors on a large number of nanoparticles follows the Poisson distribution law. This is based on a realistic consideration for ensemble measurements. For instance, when 1:1 molar ratio of NPs and molecules are randomly conjugated in solution, some of the NPs would bond with more than one dyes while some others may even have no dye bonding. This would lead to deviation between theoretical modelling and experimental observation if only the exact 1:1 binding was considered. Herein, we have considered a distribution function of $\mu^n \exp(-\mu)/n!$ in the model (see Eq. S5 in the SI), and the FRET efficiency for infinitely small NPs becomes concentration dependent (otherwise the *Eff* would

equal to 1 for all the concentration of BDP).

For the i-SET model, however, due to the complex energy migration processes we cannot derive an explicit mathematical model to describe the exact energy transfer process. Instead, we applied a phenomenological model by introducing an experimental data fitted parameter \bar{k}_{DA} (Eq. S10). In this case, we have to simplify the BDP coverage factor μ considering only the simplest case where μ molecules are binding to a nanoparticle. This is the reason we observed a deviation between the two models when the particle size is very small. Nonetheless, for particles with diameter larger than the Förster critical distance, we find that the comparison between the two models would be much reliable, because the increasing number of donor-acceptor energy transfer paths in large nanoparticle has averaged the effect of μ . We thank the reviewer for pointing out this critical concern. To avoid misleading, we have modified the Fig. S20 showing the comparison of the models in a more reliable particle size range.

4. The history of Yb-Tb upconversion is slightly longer and rich.

Response: We have found more literature on the history of Yb-Tb upconversion and cited with two represented works (*Phys. Rev. B* 1970, **1**, 4208; *Nanoscale* 2014, **6**, 1855) as ref. 35 and 36 in the revised manuscript.

5. Fig.2f - it is not clear and not explained why such behavior is observed and what it really means. More in depth discussion is needed, including the discussion about the radiative rates of A and D.

Response: Fig. 2f indicates the contribution of emission from each BDP molecules upon sensitized by FRET. The average emission intensity in Fig. 2f is calculated by the equation $I = Eff/\mu$, where μ is the attached dye number per nanoparticle and Eff is the efficiency of energy transfer calculated by comparing the luminescence lifetime changes of the NPs in the presence and absence of the acceptor molecules.

The purpose of Fig.2f is to verify the contribution of each molecules for the i-SET process. For conventional FRET processes, the energy transfer from a D to an A is normally considered as an independent event, so that the emission intensity of the A is linearly proportional to the concentration of A. For the i-SET process, as shown in Fig 2f, the average emission intensity from each BDP molecule decreases as the number of BDP molecules increases. The observation clearly indicates that the energy transfer to each BDP molecule in an i-SET process is correlated. There are competitions for each BDP molecule to accept excitation energy from the nanoparticles. This finding suggests that i-SET would be more sensitive when the acceptor concentration is lower.

As suggested, we have included a detailed discussion on the behavior observed in Fig. 2f in the revised manuscript.

“In addition, the model calculations also predict that BDP molecules would compete for accepting energy from the i-SET nanoparticle donors, as each additional molecule will reduce the probability of energy transfer to existing molecules. As shown in Fig. 2f, the emission intensity of each dye molecule decreases as the dye coverage increases. The single dye contribution was found to drop by around 60% when the coverage of dye molecules increased from 1 to 7 molecules per nanoparticle. This behaviour is different to the direct FRET in which energy transfer to a dye acceptor is independent to the addition of other molecules in the same concentration range. The results imply that the highly-doped nanoparticles will be more sensitive to small molecules at a low concentration region, which would be particularly suitable for single

molecule detection.”

6. Fig.4 - what are the expected reasons for >60 minutes photo stability when most A molecules photo bleach faster. I know this is statistics, but can the authors draw deeper conclusions about this interesting observation. Does this behavior change (and how) with excitation intensity, number of A molecules per NP? What is the environment the measurements were made in? Most biosensors require water based conditions.

Response: In our opinion, the occasional observation of molecules with extremely long photostability can be well explained by the statistics phenomenon of molecular reactions. This is analogue to the statistics decay processes such as spontaneous emission. For instance, when one counts the photon frequency of spontaneous emission for a given luminophore with luminescence lifetime of τ , it can be found that most of photons would emit within the time τ . However, there must be some excited states which are able to survive much-longer than τ . It is exactly what we observed in our experiment as we found very few of A molecules had the long photostability. We believe this is an interesting phenomenon that can only be observed by single molecule measurements. For ensemble measurements, the contribution of these long-lived molecules would be too small to be identified due to the statistics averaging.

For the following questions raised by the reviewer (the excitation intensity, effect of surface loading, and environment), at current stage we only have some preliminary results. We have changed the excitation intensity at a saturation power range of 10^5 - 10^7 W/cm² (Fig. S24). It was found that the change in excitation intensity has a negligible effect to the decay rate of the molecules when the emission of the NPs has been saturated. We did not test the NPs in water-based conditions. Currently, we only measured the NPs on cover glasses immersed with microscope immersion oil. It is kind of difficult for us to clearly explain all of the single molecule behaviors to our current knowledge. Nevertheless, we definitely agree with the reviewer that this is an interesting topic for further study. Indeed, we plan to systematically investigate the single particle phenomenon in a separate study. And we hope the reviewer concur.

7. I found numerous figures and tables in SI, which are not commented in main article (which would be desirable, so don't skip these figures, but add comments). For example - how the number of A dyes per NP has been calculated? Some more detailed information about statistics of number of A molecules per NPs should be provided and discussed.

Response: As suggested, we have added comments for all the SI figures and tables in the main article. We have also provided the detail information about how to calculate the loading concentration of A molecules per nanoparticle in the revised manuscript. It is now in page 6 and 7 of revised Supplementary Text.

8. The authors presented acceptor molecules directly anchored to the surface of NP. But actual biosensors require bio-specific molecules such as antibody-antigen or DNA molecules, which will increase the distance between superficial Tb donors to the acceptor present a few nm away (due to intermediate bio-recognition molecules). What is the Forster distance for the given Tb-BDP molecules? Did authors test any distance dependence for this interesting NPs?

Response: Due to the narrow emission bandwidth of Tb³⁺, the spectral overlapping between the Tb³⁺ emission and the BDP absorption is substantially large. As a result, we obtained a relatively longer Förster critical distance of ~6 nm for the given Tb-BDP pairs, which should make the FRET considerably efficient

even when the nanoparticles and the dye molecules are separated with a few nm by bio-specific molecules for practical applications. We have mentioned the calculated Förster critical distance in the main text.

Besides, although it is not a systematic investigation, we have tested the distance dependence of the NPs by examining the effect of particle size to the energy transfer efficiency. The results are shown in the inset of Figure 2d (see also Figure R4 below). Because of the separation of Tb^{3+} and BDP to a longer distance in the larger sized nanoparticles, for the i-SET samples the energy transfer efficiency has a moderate decrease from around 60% to 20% while increases the size of the nanoparticles from 20 to 40 nm. This result may suggest the i-SET process would have a higher resistance to the distance separation. In fact, we truly believe the distance for actual biosensors as the reviewer mentioned is of vital importance, which is also what we are exploring in our following works.

Figure R4. Energy transfer efficiency (ET Eff) plotted as a function of particle size for nanoparticles decorated with an average of 3 BDP per particle.

Some language mistakes have been found and need correction (e.g. "intensity of single BDP molecule_S_", "reaches_to_a maximum" and others).

Response: We have corrected these and other language mistakes in the revised manuscript.

Overall, I liked the concept and the quality of the presented results. I find it novel and I am sure this will be inspiration for other scientists working in the field of bio sensing and lanthanide doped applications.

Response: Once again, we appreciate the reviewer's recognition about the concept and the quality of our work, and the positive comments on the novelty and impact of the study.

Reviewer #3:

This study by Deng and coworkers demonstrates single-particle energy transfer between lanthanides in upconversion nanoparticles (UCNPs) and surface-bound BODIPY fluorophores, optimizing Ln content and demonstrating wire-like Ln chains that mediate energy transfer from the core of the nanoparticle to surface fluorophores. This is an interesting and dense study (with about 40 SI figures), and it includes some demanding single particle experiments. Overall, I would say the characterization of these nanoparticles is rigorous and interesting, particularly the impressive experiments in which the fluorophores are bleached one at a time to reveal stoichiometry. Some of the design work of how to optimize signal from the

fluorophores is similar to previous studies from the past few years, and my sense is the paper should emphasize its key strength, which is the single particle work. I think a manuscript with some revisions would make a compelling addition to the UCNP and single particle imaging literature.

Response: We thank the reviewer's positive feedback and suggestions.

Specific comments:

1) Much of the nanoparticle design work here has been explored in detail before, including similar shell structures of NaLnF₄ (JACS 141, 42 (2019)) and optimization of energy transfer to fluorescent reporters by varying Ln doping and surface dye attachment (Opt Mat 84, 345 (2018)). The latter paper also has data on enhanced fluorophore photostability, which is somewhat at odds with the current study. It would be useful to put the current findings in context of this prior work.

Response: The reviewer's point is well taken. As suggested, we have included a short discussion on these prior works in the *introduction* as follows:

“However, these measurements ... to a nanoparticle. Although considerable efforts have been devoted to improving the sensitivity and stability of FRET sensitization such as constructing confined core-shell-shell architecture and increasing the longevity of organic fluorophores^{20,21}, for quantitation of molecular targets at single-molecule levels, ...”

2) “As shown in Figure 2c, even at a low BDP concentration of ~0.5 BDP per nanoparticle, the sensitized emission from BDP molecules is still detectable with a signal-to-noise ratio of up to 30.”

Signal:noise is not the correct term for the calculation in 2c. For cuvette measurements, the noise of a fully background-subtracted spectrum is 0, so signal:noise would be infinite. More appropriate here would be a difference spectrum to see if the peak matches the free BODIPY dye emission, and to calculate the ET efficiency. This BODIPY has an unfortunately small Stokes shift, which may make calculations of ET efficiency difficult.

Response: We apologize for the misunderstanding caused by the unclear interpretation about the signal-to-noise ratio. The discussion of sensitized emission of samples with low BDP loading concentration is aimed to indicate high detection limit of our system rather than calculate the ET efficiency (the study of ET efficiency was presented in Figure 2d). We agree with the reviewer that for cuvette measurements the background spectrum can be subtracted. However, even with a full subtraction the background noise is still not completely 0. There is a small fluctuation in the baseline spectrum due to the instrument response. For example, in a typical case of measurement we get a background fluctuation of about ±20 counts (see Figure R5 below). Herein, in order to obtain a meaningful signal from the sensitized emission, the detectable emission intensity of dye molecules must be at least 3-10 times higher than the fluctuation of the background spectrum. In this experiment, we observed the sensitized emission of BDP is over 30 times higher than the background spectrum even when the loading concentration of BDP is as low as 0.5 per nanoparticle. We believe this is a good indication implying that our system can work for very low detection limit. Nevertheless, we have noted that the detection limit of the instrument is correlated to the molar concentration of dye molecule in solution. For clarity, we have amended the discussion in the manuscript as follows:

“As shown in Figure 2c, the sensitized emission signal from the BDP molecules is around 30 times higher

than the background spectrum ($S/N = 30$) even when the loading concentration is as low as ~ 0.5 BDP per nanoparticle (corresponding to ~ 500 nM BDP in solution). The result suggests the possibility of achieving highly sensitive FRET detection by adopting *i*-SET.”

Figure R5. Luminescence spectrum showing the baseline fluctuation of the fluorescence spectrometer after fully subtraction of the background spectrum.

3) The discussion of *i*-SET (a new term?) and cartoons in Fig 1 leave the impression of direct-contact energy transfer, such as Dexter e^- transfer. I don't think this is likely, nor is there evidence for this. Even if an *i*-SET Ln wire ET process moves energy from the nanocrystal interior toward the surface, ET between Ln and fluorophore is still most likely a FRET process. The authors might consider clearer ways to distinguish FRET from the core vs FRET from the shell. If there is evidence of non-FRET processes such as Dexter, the authors should highlight it.

Response: We agree with the reviewer that ET between Ln and fluorophore should be a FRET process. However, we get some clues (although it is still not conclusive) that the intraparticle Tb^{3+} - Tb^{3+} energy transfer is governed by a Dexter-type exchange mechanism. This is supported by our model simulation in Eq. S15 and Fig. S14, in which only if exchange interaction between Tb^{3+} was considered the experimental data could fit well with the calculated \bar{k}_{DA} . It is also supported by literature reports where people have demonstrated that energy migration between lanthanides in many host materials was governed by exchange interaction (see e.g. *J. Lumin.* 1988, **42**, 275-282; *Appl. Phys. Lett.* 1966, **9**, 255-256). Therefore, the intraparticle energy transfer in the proposed *i*-SET process can be either DET or FRET or both. We agree that the *i*-SET scheme in Figure 1 leaves an impression that Tb^{3+} to BDP is direct-contact energy transfer. To avoid misleading, we have amended Figure 1 highlighting different energy transfer processes from the nanoparticles. The revised figure is also shown as Figure R6 below:

Figure R6. Schematic of single upconversion nanoparticle-based molecular sensing. The sensing of individual molecules is realized by probing FRET signal of nanoparticle-to-molecule energy transfer. The i-SET path shows energy transfer from nanoparticles with a high content of activators (represented as cyan spheres) in which excitation energy can fast migrate within the particles by either Dexter energy transfer (DET) or FRET and lead to promoted sensitization of adhesive molecules through intraparticle-to-surface FRET. The direct ET path indicates energy transfer from conventional upconversion nanoparticles having low doping concentration of activators in which activators buried inside the nanoparticle can hardly transfer their energy to the molecular acceptors due to the longer distance of separation.

4) Given the direct contact between BODIPY and heavy atom shell, as well as rapid O₂-mediated quenching, do the authors believe triplets are forming in the BODIPY? There is evidence of this in NIR dye sensitization of Yb/Er UCNPs, although this remains somewhat controversial.

Response: The reviewer has provided an insightful thought of the study. In view of the fact that the formation of triplets are a major path for photo-blinking and bleaching of BODIPY molecules, we can say for sure that triplets are forming in our system. However, at this moment it is hard to verify if the heavy atom shell promotes the formation of triplet states of BODIPY or not. As the study of triplets is already out of the scope of this work, we believe it will be an interesting topic for further investigation in the future.

5) “Nevertheless, for conventional Er³⁺, Tm³⁺, or Ho³⁺ activators co-doped with Yb³⁺ or Nd³⁺ ions as sensitizers²¹⁻²³, a high content of activator and sensitizer combination would generally lead to low luminescence efficiencies of the nanoparticles due to concentration quenching effects²⁴.”

There is a growing literature at this point on high-Ln UCNPs, and it has become clear that concentration quenching is not a significant factor in core/shell UCNPs. The authors seem to recognize that adding Yb-free shells necessarily cuts off the prime energy loss pathway. See e.g., JACS 139, 3275 (2017) and Nat Comm 9, 3082 (2018) for previous, in-depth discussions of high Ln core/shell UCNPs and why the concept of concentration quenching is dated. I think the discussion in the current paper can take this prior work into account to substantially shorten this part of the manuscript.

Response: We thank the reviewer for the useful references. As suggested, we have cited these two papers as references 26 and 31 in the revised manuscript, and shorten the description about the concentration

quenching in *introduction* as follows:

“Nevertheless, for conventional Er^{3+} , Tm^{3+} , or Ho^{3+} activators co-doped with Yb^{3+} or Nd^{3+} ions as sensitizers²²⁻²⁴, a high content of activator and sensitizer combination would generally lead to low luminescence efficiencies of the nanoparticles due to concentration quenching effects²⁵⁻³¹. It mainly conducts through coupled energy transfer from the lower intermediate states of sensitizers/activators to –OH and –CH_n vibration modes of surface molecules/defects (Supplementary Scheme. S1 and 2).”

REVIEWER COMMENTS

Reviewer #1 (Remarks to the Author):

The point-by-point response to the referees comments is satisfactory and so the revised manuscript is recommended for publication in Nature Communications.

Reviewer #2 (Remarks to the Author):

The article has been corrected in response to reviewers comments. Despite I have some further questions, these questions do not change my opinion about high quality of this research and opinion this article is suitable for Nature Communication.

1. Concerning my previous Q3 about Figure S20 - I understand that the models for direct FRET and i-FRET require differences, but still I want to know more, what this graph is teaching us. For example the direct FRET in YbEr with Y only shell (the YbEr@Y NPs) will depend from Y shell thickness apart from NP diameter - so the presented 'picture' of phenomenon is not complete. Moreover, in Fig.S20b, the ET Efficiency decreases for rising size of NP because of increasing the surface area of a NP for a given amount of acceptor - kind of "diluting" the acceptors per NP surface (ET Eff/ μ) and decreasing the relative contribution of Tb emission being quenched by acceptor to the total of all emitting Tb ions - is that what the model suggests? What is then the optimum balanced NP size, when one takes into account the brightness of NPs (which is rising for rising NP diameter, I guess) vs decreasing FRET efficiency. I think this is important to use a reliable model to predict/compare the properties of the system, but here more critical discussion is then needed.

2. Concerning Fig.2f - Taking into account the that fact the multiple superficial donor ions (either Er or Tb) may transfer the energy to a single acceptor molecule (many-to-one or many-to-few), and the fact the lifetimes of both lanthanides are in 0.1-1 ms range, while that of acceptor is in the ~ 1 ns range, I would say, that in both cases, the acceptor brightness shall saturate only for larger A quantities, but not for single acceptor molecules, as many multiple-Ln-single-BDP FRET events have a chance to take place within the average lifetime of excited state on Ln. Approximately up to $0.1-1\text{ms}/1\text{ns} \sim 10^5-10^6$ times (per lifetime of excited state of Ln³⁺) the FRET may occur between D and A. Then, when acceptor receives the energy from donor, returns (within single ns) to ground state (emitting photons) and becomes ready for the next FRET event. Therefore, in my opinion, the brightness of acceptor should not (for low concentrations of acceptors) depend (decrease here - Fig.2f) on its concentration in these circumstances. Can the authors comment on this? Can you support your modelling with experimental data in this area?

Concerning the suggestion by the reviewer #3 - Dexter energy (or Dexter electron) transfer is a fluorescence quenching, in which excited electron of the donor is transferred to acceptor via non-radiative way. Firstly, this happens typically within short 1 nm distance, secondly, optically active electrons of the donor (here Tb³⁺) are f-electrons, which do not participate in chemical reactions and are covered/protected by higher laying orbitals, thus I am not sure, that using DET to explain the ET here is correct. Tb³⁺ is one of rare earth ions that may change the oxidation state (to +4), indeed, but it seems it does not apply here.

Reviewer #3 (Remarks to the Author):

I like this paper and offered a generally positive review along with some specific suggestions where I thought the paper needed improvement. I'm surprised to see a couple of those suggestions were not addressed in the paper, as these do not involve new experiments. I think

this paper should be published, after dealing with these specifics.

For #2, the authors should add in a figure with difference spectra (ie, between spectra with and without the fluorophores). I would like to see the isolated BODIPY peaks. Are the peaks where the BODIPY peaks are if the fluorophore is excited directly, or is there some other electronic effect that shifts the peak for the fluorophores on the nanoparticle surface? That was the key point.

I've not seen a signal:noise ratio used for a cuvette measurement before, and I think it is odd to include it. S/N is useful in microscopy because highly localized signal (eg, an individual pixel) is often difficult to attribute to a specific source in the absence of integration across a large sample volume like a cuvette. If the authors require a S/N measurement even in a cuvette, this makes me less excited about the current results. I do not think it helps them to highlight this.

For #5, my point was that recent work (the references I suggested) has shown that the concept "concentration quenching" is sometimes unimportant for core/shell UCNP structures. Yet the authors add these references to a sentence about concentration quenching, a concept those references specifically do not support.

I would also double check the references, as I notice some of the names are incorrect.

2) "As shown in Figure 2c, even at a low BDP concentration of ~ 0.5 BDP per nanoparticle, the sensitized emission from BDP molecules is still detectable with a signal-to-noise ratio of up to 30." Signal:noise is not the correct term for the calculation in 2c. For cuvette measurements, the noise of a fully background-subtracted spectrum is 0, so signal:noise would be infinite. More appropriate here would be a difference spectrum to see if the peak matches the free BODIPY dye emission, and to calculate the ET efficiency. This BODIPY has an unfortunately small Stokes shift, which may make calculations of ET efficiency difficult.

Response: We apologize for the misunderstanding caused by the unclear interpretation about the signal-to-noise ratio. The discussion of sensitized emission of samples with low BDP loading concentration is aimed to indicate high detection limit of our system rather than calculate the ET efficiency (the study of

ET efficiency was presented in Figure 2d). We agree with the reviewer that for cuvette measurements the

background spectrum can be subtracted. However, even with a full subtraction the background noise is still not completely 0. There is a small fluctuation in the baseline spectrum due to the instrument response. For example, in a typical case of measurement we get a background fluctuation of about ~ 20 counts (see Figure R5 below). Herein, in order to obtain a meaningful signal from the sensitized emission, the detectable emission intensity of dye molecules must be at least 3-10 times higher than the fluctuation of the background spectrum. In this experiment, we observed the sensitized emission of BDP is over 30 times higher than the background spectrum even when the loading concentration of BDP is as low as 0.5 per nanoparticle. We believe this is a good indication implying that our system can work for very low detection limit. Nevertheless, we have noted that the detection limit of the instrument is correlated to the molar concentration of dye molecule in solution. For clarity, we have amended the discussion in the manuscript as follows:

"As shown in Figure 2c, the sensitized emission signal from the BDP molecules is around 30 times higher than the background spectrum ($S/N = 30$) even when the loading concentration is as low as ~ 0.5 BDP per nanoparticle (corresponding to ~ 500 nM BDP in solution). The result suggests the possibility of achieving highly sensitive FRET detection by adopting i-SET."

5) "Nevertheless, for conventional Er^{3+} , Tm^{3+} , or Ho^{3+} activators co-doped with Yb^{3+} or Nd^{3+}

ions as sensitizers²¹⁻²³, a high content of activator and sensitizer combination would generally lead to low

luminescence efficiencies of the nanoparticles due to concentration quenching effects²⁴.”

There is a growing literature at this point on high-Ln UCNPs, and it has become clear that concentration

quenching is not a significant factor in core/shell UCNPs. The authors seem to recognize that adding Yb-free shells necessarily cuts off the prime energy loss pathway. See e.g., JACS 139, 3275 (2017) and Nat Comm 9, 3082 (2018) for previous, in-depth discussions of high Ln core/shell UCNPs and why the concept of concentration quenching is dated. I think the discussion in the current paper can take this prior work into account to substantially shorten this part of the manuscript.

Response: We thank the reviewer for the useful references. As suggested, we have cited these two papers

as references 26 and 31 in the revised manuscript, and shorten the description about the concentration quenching in introduction as follows:

“Nevertheless, for conventional Er³⁺, Tm³⁺, or Ho³⁺ activators co-doped with Yb³⁺ or Nd³⁺ ions as sensitizers²²⁻²⁴, a high content of activator and sensitizer combination would generally lead to low luminescence efficiencies of the nanoparticles due to concentration quenching effects²⁵⁻³¹. It mainly conducts through coupled energy transfer from the lower intermediate states of sensitizers/activators to –OH and –CH_n vibration modes of surface molecules/defects (Supplementary Scheme. S1 and 2).”

Reviewer #1 (Remarks to the Author):

The point-by-point response to the referee's comments is satisfactory and so the revised manuscript is recommended for publication in Nature Communications.

Response: We thank the reviewer's recognition and their comments from the previous round that have greatly strengthened the manuscript.

Reviewer #2 (Remarks to the Author):

The article has been corrected in response to reviewers comments. Despite I have some further questions, these questions do not change my opinion about high quality of this research and opinion this article is suitable for Nature Communication.

Response: We thank the reviewer for the comments. Detailed responses to the remaining concerns are found below.

1) 1. Concerning my previous Q3 about Figure S20 - I understand that the models for direct FRET and i-FRET require differences, but still I want to know more, what this graph is teaching us. For example the direct FRET in YbEr with Y only shell (the YbER@Y NPs) will depend from Y shell thickness apart from NP diameter - so the presented 'picture' of phenomenon is not complete. Moreover, in Fig.S20b, the ET Efficiency decreases for rising size of NP because of increasing the surface area of a NP for a given amount of acceptor - kind of "diluting" the acceptors per NP surface (ET Eff/ μ) and decreasing the relative contribution of Tb emission being quenched by acceptor to the total of all emitting Tb ions - is that what the model suggests? What is then the optimum balanced NP size, when one takes into account the brightness of NPs (which is rising for rising NP diameter, I guess) vs decreasing FRET efficiency. I think this is important to use a reliable model to predict/compare the properties of the system, but here more critical discussion is then needed.

Response: We thank the reviewer for the critical comments. For the model calculation in Figure S20, we have considered the core-shell structures by assuming a v/v ratio of 1:1 between core and shell layers (e.g. the core size of 7.9 nm for a 10 nm core-shell nanoparticle, and 15.9 nm for a 20 nm nanoparticle), which is close to the real situations in our experiments. In addition, the efficiency decreases for rising size of NP is mainly due the distance separation rather than diluting the acceptors per NP surface. For clarity, we have conducted a new set of calculations in which the BDP coverages were normalized to the same concentration per NP surface (see **Figure R1** below). It can be seen that the ET efficiency is still monotonically decreasing as the particle size increases. This is because the increase in particle size separates the active dopants (particularly for those buried deep inside the nanoparticles) away from the surface anchored molecular acceptors. Even with a high coverage of the acceptors on the surface, a larger particle would still have lower energy transfer efficiency, since the active lanthanides deep inside the nanoparticles are already far away from the surface and are out of the critical distance for efficient FRET.

Nonetheless, as correctly pointed out by the reviewer, the conclusions in Figure S20 are only based on the model calculations which lacks of critical discussion. To avoid any possible misleading, we decided to delete the Fig. S20 and skipped the discussion of size effects in this work, since at this moment we don't have enough experimental data to validate the reliability of this part of modelling. This does not influence

other model calculations in the main text, in which the model calculations can fit well with the experimental findings.

Figure R1. Plots of calculated energy transfer efficiency versus the diameter of nanoparticles with different dye coverage.

2) 2. Concerning Fig.2f - Taking into account the fact that the multiple superficial donor ions (either Er or Tb) may transfer the energy to a single acceptor molecule (many-to-one or many-to-few), and the fact that the lifetimes of both lanthanides are in the 0.1-1 ms range, while that of the acceptor is in the ~ 1 ns range, I would say that in both cases, the acceptor brightness shall saturate only for larger A quantities, but not for single acceptor molecules, as many multiple-Ln-single-BDP FRET events have a chance to take place within the average lifetime of the excited state on Ln. Approximately up to $0.1\text{-}1\text{ms}/1\text{ns} \sim 10^5\text{-}10^6$ times (per lifetime of the excited state of Ln³⁺) the FRET may occur between D and A. Then, when the acceptor receives the energy from the donor, it returns (within single ns) to the ground state (emitting photons) and becomes ready for the next FRET event. Therefore, in my opinion, the brightness of the acceptor should not (for low concentrations of acceptors) depend (decrease here - Fig.2f) on its concentration in these circumstances. Can the authors comment on this? Can you support your modelling with experimental data in this area?

Response: We understand the reviewer's point. As the organic acceptors can very efficiently consume excitation energy from multiple Ln donors, the brightness of these acceptors can be hardly saturated at low concentration. Nevertheless, we need to clarify that the decrease in the brightness of acceptors in Fig. 2f is not due to the saturation of the acceptor's emission but the insufficient energy donation from the lanthanide donors ----- the existing acceptors are so efficient in consuming the excitation energy from a large quantity of donors so that the additional acceptor cannot find enough energy for emission. It should be noted that the y-axis in Fig. 2f is not the total emission brightness of multiple BDPs but an average brightness of each BDP when a certain amount of the molecules (number on the x-axis) were bonded to a nanoparticle.

As an experimental evidence, in figure 2d, we found that loading 1 BDP per NP can lead to an energy transfer efficiency of 25%, meaning that a single BDP molecule is capable of consuming about 1/4 of the excitation energy from a NP through i-SET. The energy transfer efficiency increases to 54% when the number of BDP per NP increases from 1 to 4 ----- the accepted energy for each BDP, herein, dropped from 25% to $54/4=13.5\%$ indicating that the energy supply from the donors to each molecule becomes insufficient even at such a low acceptor concentration. We believe this is an important finding indicated

that by applying i-SET approach it is not necessarily needed to use multiple acceptors to maximize FRET signal which offers opportunity for single molecule detection. For a better comparison, we have modified the Fig. 2f including experimental data along with the calculation results (see also **Figure R2** below).

Figure R2. Simulated and experimental average luminescence intensity of single BDP molecule as a function of BDP coverage on individual nanoparticles.

3) Concerning the suggestion by the reviewer #3 - Dexter energy (or Dexter electron) transfer is a fluorescence quenching, in which excited electron of the donor is transferred to acceptor via non-radiative way. Firstly, this happens typically within short 1 nm distance, secondly, optically active electrons of the donor (here Tb³⁺) are f-electrons, which do not participate in chemical reactions and are covered/protected by higher laying orbitals, thus I am not sure, that using DET to explain the ET here is correct. Tb³⁺ is one of rare earth ions that may change the oxidation state (to +4), indeed, but it seems it does not apply here.

Response: This is a very critical concern. Indeed, it is also a question that we are bothering for a very long time. We agree with the reviewer that DET seems not applicable to explain 4f energy transition between trivalent lanthanides. However, as we mentioned in the previous response to the Reviewer #3, we did observe some experimental results which are deviated from dipole-dipole FRET.

To be brief, we have quantitatively analyzed the effect of Tb³⁺ concentration in the shell layer of NaYbF₄:Tb@NaLuF₄:Tb core-shell nanoparticles to energy transfer from the NPs to the BDP molecules. As shown in Fig. S14 in SI and **Figure R3** below, we observed an “S” shape change in NP-BDP energy transfer constant \bar{k}_{DA} as a function of the Tb³⁺ concentration. The experimental data cannot be fitted with our model equation if only dipole-dipole energy transfer (r^{-6} dependent rate evolution) was considered. In particular, the data show a clear concentration threshold of >30 mol% Tb³⁺ (equivalent to a calculated average Tb³⁺-Tb³⁺ distance of < 6 Å) for the core-shell nanoparticles to initiate efficient energy transfer. We found that 10-30 mol% Tb³⁺ contributes almost negligible to the \bar{k}_{DA} , which cannot be explained by the classical FRET theory, because \bar{k}_{DA} should monotonously increase rather than become an “S” shape if it is dipole-dipole energy transfer.

Notably, we have found some of the previous studies which also reported similar results suggesting short-range exchange interaction for systems with high concentration of lanthanides, for example please see: for Gd-Gd energy migration, *J. Lumin.* 1986, *35*, 155; and more recently for Tb-Tb-Ce energy transfer, *J. Phys. Chem. C* 2011, *115*, 3475. However, this is still under debate, as very recently we happened to

notice that some researchers argue that the dipole-dipole approximation may break down and higher-order multipoles may contribute more in short donor-acceptor distances (*Int. J. Quantum Chem.* 2014, 114, 102). Because the validation of energy migration mechanism is not the central goal of this work, we would like to leave it as an open question for further investigation. Nevertheless, we still wish to keep the description about DET in Figure 1. As it is a schematic showing the general design of the process, we wish to make it a broader concept and can perhaps inspire more people working in different area of materials (e.g. Ln-MOF and QDs). We hope the reviewer concur after go through our clarification.

Figure R3. Concentration dependence of the fit parameter \bar{k}_{DA} of the nanoparticle-fluorophore energy transfer.

Reviewer #3 (Remarks to the Author):

I like this paper and offered a generally positive review along with some specific suggestions where I thought the paper needed improvement. I'm surprised to see a couple of those suggestions were not addressed in the paper, as these do not involve new experiments. I think this paper should be published, after dealing with these specifics.

1) For #2, the authors should add in a figure with difference spectra (ie, between spectra with and without the fluorophores). I would like to see the isolated BODIPY peaks. Are the peaks where the BODIPY peaks are if the fluorophore is excited directly, or is there some other electronic effect that shifts the peak for the fluorophores on the nanoparticle surface? That was the key point. I've not seen a signal:noise ratio used for a cuvette measurement before, and I think it is odd to include it. S/N is useful in microscopy because highly localized signal (eg, an individual pixel) is often difficult to attribute to a specific source in the absence of integration across a large sample volume like a cuvette. If the authors require a S/N measurement even in a cuvette, this makes me less excited about the current results. I do not think it helps them to highlight this.

Response: We apologize for not properly addressing some specific comments in the previous round. We greatly thank the reviewer's suggestions, which have helped to improve the manuscript significantly. We have worked to clarify the remaining points raised, as detailed below

For the response #2, we now understand the reviewer’s point. we have provided a difference spectrum between spectra with and without the fluorophores, and compared it with fluorescence peaks of pristine BDP, as shown in the figure below (**Figure R4**). We do not find obvious shift in BDP’s emission suggesting the absence of significant electronic effects to the BDP on the nanoparticle surface. As suggested, we have included this figure in the revised Figure 2c and replaced the corresponding description about S/N in the main text to

“As shown in Figure 2c, the sensitized emission of BDP can be clearly detected even when the loading concentration is as low as ~0.5 BDP per nanoparticle. The difference emission spectrum of nanoparticles with and without the BDP molecules (Figure 2c inset) indicates that the sensitized emission is almost identical to the downshifting emission of BDP through direct excitation. It confirms that the properties of the fluorophores are well retained at the nanoparticle’s surface.”

Figure R4. Upconversion luminescence spectra of NaYbF₄:Tb(40 mol%)@NaTbF₄ nanoparticles with and without an average of 0.5 BDP loading per nanoparticle. The inserted diagram shows a comparison between the direct emission spectrum of free BDP molecules at 365 nm excitation and the sensitized emission spectrum of coupled BDP obtained by calculating the spectral differences between emission spectra of the nanoparticles with and without 0.5 BDP conjugation.

2) For #5, my point was that recent work (the references I suggested) has shown that the concept “concentration quenching” is sometimes unimportant for core/shell UCNP structures. Yet the authors add these references to a sentence about concentration quenching, a concept those references specifically do not support.

Response: We have made a substantial revision to the last paragraph of *Introduction* taking into account recent progresses on adopting core-shell structures to minimize the effect of concentration quenching. The revised description is as followed

“This process can significantly enhance fluorescence signals ... at a single-particle level. Nevertheless, for conventional Er³⁺, Tm³⁺, or Ho³⁺ activators co-doped with Yb³⁺ or Nd³⁺ ions as sensitizers²²⁻²⁴, a high content

of activator and sensitizer combination would generally lead to low luminescence efficiencies due to the concentration quenching effect²⁵⁻²⁷. In view of recent advances on using sensitizer-free shell coating to block prime concentration quenching paths via energy migration to surface quenchers²⁸⁻³¹, we envisage that efficient i-SET can be achieved by utilizing activator shell layers disfavoring energy transfer to the lower lying quenching sites such as –OH and –CH_n vibration modes (Supplementary Scheme. S1 and 2).”

I would also double check the references, as I notice some of the names are incorrect.

Response: We are sorry for the mistakes. We have carefully checked the references and made correction to all the mistakes.

In closing, the authors wish to thank all of the reviewers for comments, questions and suggestions. The article has been carefully checked and revised. All changes were marked with red color in the article and SI. We hope that the reviewer now finds our work suitable for publication

REVIEWERS' COMMENTS:

Reviewer #2 (Remarks to the Author):

Thank You for clarifications and comments. Herein I conclude the manuscript can be accepted for publication in Nature Communication.

Short comments of mine, with no influence on the above decision and with no need to respond.

1. It is a pity the authors have decided to remove Figure S20, as this discussion and forward looking thinking is valuable
2. The DEX discussion is really interesting. I suggest also to consider non homogenous distribution of Tb³⁺ ions in the NP volume, which we found to be responsible for some spectral nuances in our own studies.

Reviewer #3 (Remarks to the Author):

All of my concerns have been addressed. Nice paper.

Reviewer #2 (Remarks to the Author):

Thank You for clarifications and comments. Herein I conclude the manuscript can be accepted for publication in Nature Communication. Short comments of mine, with no influence on the above decision and with no need to respond.

Response: We are grateful for the reviewer's favorable comments.

1. It is a pity the authors have decided to remove Figure S20, as this discussion and forward looking thinking is valuable

Response: We are sorry that we decided to remove Figure S20 at this moment. We are also willing to discuss this model in the future when we are able to evaluate its reliability with more experimental results.

2. The DEX discussion is really interesting. I suggest also to consider non homogenous distribution of Tb^{3+} ions in the NP volume, which we found to be responsible for some spectral nuances in our own studies.

Response: Thank you very much for the suggestion. We will consider the in-homogenous factors in our further studies.

Reviewer #3 (Remarks to the Author):

All of my concerns have been addressed. Nice paper.

Response: We thank the reviewer's comments and suggestions which have greatly improved the work.